# PGPB Isolated from Drought-Tolerant Plants Help Wheat Plants to Overcome Osmotic Stress

**DOI:** 10.3390/plants13233381

**Published:** 2024-11-30

**Authors:** Veronika N. Pishchik, Elena P. Chizhevskaya, Vladimir K. Chebotar, Galina V. Mirskaya, Yuriy V. Khomyakov, Vitaliy E. Vertebny, Pavel Y. Kononchuk, Dmitriy V. Kudryavtcev, Olga A. Bortsova, Nina G. Lapenko, Igor A. Tikhonovich

**Affiliations:** 1All-Russia Research Institute for Agricultural Microbiology, Podbelskogo Hwy 3, Pushkin, 196608 Saint Petersburg, Russia; chizhevskaya@yandex.ru (E.P.C.); dv_kudryavtsev@arriam.ru (D.V.K.); bortsova17@yandex.ru (O.A.B.); i.tikhonovich@arriam.ru (I.A.T.); 2Agrophysical Scientific Research Institute, Grazhdanskiy pr. 14, 195220 Saint Petersburg, Russia; galinanm@gmail.com (G.V.M.); himlabafi@yandex.ru (Y.V.K.); verteb22@mail.ru (V.E.V.); 79117717774@yandex.ru (P.Y.K.); 3North Caucasus Federal Agrarian Research Centre, Zootechnical Lane, 15, 355017 Stavropol, Russia; lapenko31@yandex.ru; 4Department of Genetics and Biotechnology, Faculty of Biology, Saint Petersburg State University, 7-9 Universitetskaya Embankment, 199034 Saint Petersburg, Russia

**Keywords:** drought-tolerant plants, PGPB, wheat, phytohormones, osmotic stress

## Abstract

The aim of this research was to study the effect of plant-growth-promoting bacteria (PGPB) isolated from the drought-tolerant plants camel thorn (*Alhagi pseudoalhagi* (M.Bieb.) Fisch) and white pigweed (*Chenopodium album* L.) on wheat (*Triticum aestivum* L.) plants cv. Lenigradskaya 6, growing under hydroponic conditions and osmotic stress (generated by 12% polyethylene glycol-6000 (PEG)). Based on the assumption that plants create a unique microbiome that helps them overcome various stresses, we hypothesized that bacteria isolated from drought-tolerant plants may assist cultivated wheat plants in coping with drought stress. PGPB were isolated from seeds and leaves of plants and identified as *Bacillus* spp. (strains Cap 07D, Cap 09D, and App 11D); *Paenibacillus* sp. (Cap 286); and *Arthrobacter* sp. (Cap 03D). All bacteria produced different phytohormones such as indole acetic acid (IAA), abscisic acid (ABA), and gibberellic acid (GAS_3_) and were capable of stimulating wheat growth under normal and osmotic stress conditions. All PGPB reduced the malondialdehyde (MDA) content, increased the total chlorophyll content by increasing chlorophyll *a*, and modulated wheat hormone homeostasis and CAT and POX activities under osmotic conditions. Selected strains can be promising candidates for the mitigating of the drought stress of wheat plants.

## 1. Introduction

Drought stress affects 45% of the total arable land, and the global aridity stress area consists of 27% of the world’s agricultural areas [1,2,3].

Soft wheat (*T. aestivum*) is an important cereal crop cultivated in many regions of the world [4,5,6]. Wheat yield losses due to drought have been recorded in many countries and can be as high as 50% [7,8]. Drought damages various cellular compartments, inactivates enzymes, leads to protein degradation, reduces transpiration, and, eventually, leads to the wilting and desiccation of plants [9,10]. The following mechanisms are involved in drought: intracellular signal transduction (via serine/threonine-protein kinase); ABA (abscisic acid) signaling, together with the activation of the salicylic acid signaling pathway; the modulation of the antioxidant defense mechanism; the maintenance of the cellular energy balance; the closure of the stomata; and the regulation of the cell wall [11,12]. This stress-related process involves miRNAs which inhibit the translation of various target genes, mostly comprising transcription factors [13,14]. Candidate genes associated with adaptive gene networks in wheat to drought stress conditions have now been identified [15,16,17,18].

Plant hormonal signaling plays an important role in the process of plant adaptation to drought. Key genes have been identified that regulate ABA signaling, as well as those related to the signaling pathways of other hormones such as auxin, ethylene, jasmonic, salicylic, and gibberellic acids [18]. ABA-dependent genes associated with reactive oxygen species (ROS) inactivation and water use efficiency are involved in signal transduction and transcription pathways [19]. In general, an imbalance between ROS generation and antioxidant defense systems leads to an excessive ROS accumulation and induces oxidative stress in plants [20]. Under oxidative stress, lipid peroxidation occurs, nucleic acids and proteins are damaged, and the carbohydrate metabolism is altered [20,21]. Thus, ROS are signal transduction molecules that control various pathways in the process of plant acclimation under stress conditions [22].

Such antioxidant enzymes as peroxidase (POD) and ascorbate peroxidase (APX), catalase (CAT) and superoxide dismutase (SOD), and non-enzymatic components—proline and carotenoids—are involved in protecting plants from drought stress [23,24].

The intensity of drought is projected to increase consistently [25], hence the need for research to address this issue to ensure food security. One aspect of solving this problem, along with the use of drought-tolerant wheat genotypes [26], is the use of biologicals that help to increase the adaptive potential and yield of plants under changing environmental conditions [27].

It is known that the positive effect of PGPB on host plants under drought stress is realized due to (1) changes in enzyme activity and the induction of systemic stress resistance reactions in plants [28,29,30]; (2) the production of a wide range of biologically active compounds including hydrolytic enzymes [31], plant hormones [27], ACC deaminase [32], and exopolysaccharides [33]; (3) changes in the hormonal background of plants [34,35]; (4) changes in the root architecture and the increased availability of macro- and micronutrients [36]; and (5) the regulation of photosynthesis [12,34]. However, the efficacy of the same PGPB strain under drought stress can vary depending on the plant genotype and environment conditions [34,37,38].

It recently has been found that PGPB helped wheat plants to alleviate drought stress [39,40,41,42,43,44]. For example, the endophytic PGPB *Pantoea alhagi* isolated from the drought-tolerant plant *Alhagi sparsifolia* Shap. (*Leguminosae*) improved growth and drought tolerance in wheat [39].

Despite the current understanding of plant exposure mechanisms, some issues of wheat–microbial relationships under drought stress remain relevant for research. Different responses in the manifestation of wheat antioxidant enzyme activity in response to bacterial inoculation under drought conditions are described [28,44]. However, the bacterial mechanisms affecting endogenous plant hormone levels are not fully understood [34,45].

It is known that plants growing in different ecological niches form their own unique microbiome, which allows plants to tolerate various unfavorable conditions [46,47]. Drought-tolerant plants grow in an arid zone characterized by insufficient atmospheric moisture and high air temperatures that limit plant growth. Therefore, we hypothesized that bacteria isolated from the seeds and leaves of drought-tolerant plants could help other plants to overcome drought conditions.

This work aimed to study the effects of epiphytic bacteria isolated from the seeds and leaves of drought-tolerant plants on the growth, hormonal status, and antioxidant ferments activities of wheat plants under osmotic stress generated by 12% PEG6000.

## 2. Results

### 2.1. Bacterial Isolation

All the PGPB studied were isolated from the drought-resistant plants. The strains Cap 03 D, Cap 07D, and Cap 09D were isolated from the surface of the seeds of *C. album*, Cap 286 was isolated from the surface of the leaves of *C. album*, and App 11D were isolated from the surface of the seeds of *A. pseudoalhagi.*

### 2.2. Bacterial Identification

Most of the isolates (Cap 07D, Cap 09D, and App 11D) were identified as *Bacillus* spp. The strain Cap 286 was identified as *Paenibacillus* sp., and the strain Cap 03D was identified as *Arthrobacter* sp. For each of the strains, representatives with the closest 16S rDNA sequences were identified in the GenBank database (Appendix A). The phylogenetic analysis is presented in Figure 1.

Cap 03D, belonging to the genus *Arthrobacter*, was found to be in the same cluster with strains of *Arthrobacter agilis, A. ruber*, and *A. pityocampae* when compared with reference strains. Strains Cap 07D and Cap 09D (members of the genus *Bacillus)* are in the same cluster as strains of the *B. velezensis, B. safensis*, and *B. australimaris* species. Strain App 11D joined the cluster with *B. pumilus* and *B. zhangzhouensis*. Strain Cap 286, a member of the genus *Paenibacillus,* has merged into one cluster with *P. nicotianae* and *P. chinensis*.

### 2.3. The Effect of Temperature of the Growth of PGPB

The ability of PGPB to grow at different temperatures is presented in Table 1. Most strains were able to grow in the range of 10–40 °C, except the strain Cap 03D *Arthrobacter* sp. The maximal growth temperature of Cap 03D was 35 °C. The strain Cap 07D *Bacillus* sp. was able to grow at 5 °C. The most heat-resistant strains were strains Cap 09D and App 11D *Bacillus* spp., which were able to grow at 50 °C. Moreover, the strain Cap 09D *Bacillus* sp. was found to have weak growth at a temperature of 55 °C.

### 2.4. The Effect of Different Concentrations of NaCl on Growth of PGPB

The ability of the isolates to grow at different concentrations of NaCl is present in Table 2. All strains were able to grow on a GMF medium with 1% NaCl. Strains Cap 07D, Cap 09D, and App 11D *Bacillus* spp. were able to grow in a medium with 7% NaCl. Only two strains, Cap 09D and App 11D, grew on a medium with 10% NaCl.

### 2.5. Metabolic Activities of PGPB

All strains were tested on different metabolic activities (Table 3). The strain Cap 03D *Arthrobacter* sp. and strains Cap 07D, Cap 09D, and Cap 286 *Bacillus* spp. possessed amylolytic activity. The highest amylolytic activity was detected in the strain Cap 286 (Figure 2).

Of all the strains studied, only strain Cap 09D *Bacillus* sp. possessed lipolytic activity. Phosphate-mobilizing activity was found only in the Cap 07D *Bacillus* sp. strain, whereas proteolytic activity was detected in almost all strains with the exception of Cap 03D *Arthrobacter* sp. Moreover, the strain Cap 286 *Bacillus* sp. had cellulolytic and nitrogen-fixing activities in contrast to the other strains studied (Figure 2). The strains Cap 07D and Cap 09D *Bacillus* spp. produced NH_4_. All strains produced indoles and siderophores. Polysaccharides were revealed in all the studied strains with the exception of Cap 03D *Arthrobacter* sp. (Appendix A).

### 2.6. Phytohormone Production Activity of PGPB

We analyzed the indole activity of the tested strains when grown on the R2A medium with tryptophan supplementation (Figure 3). All bacteria studied produced indoles. The highest indole activity was found for strains Cap 07D *Bacillus* sp. (58.8 mg L^−1^) and Cap 286 *Paenibacillus* sp. (31.9 mg L^−1^), and the lowest was found for strain Cap 03D *Arthrobacter* sp. (2.8 mg L^−1^).

We have studied the ability of strains to produce such phytohormones as gibberellin (GAS_3_), indole acetic acid (IAA), and abscisic acid (ABA) by using high-performance liquid chromatography–mass spectrometry (HPLC-MC) (Table 4). All strains studied were capable of producing phytohormones such as GAS_3_ (13.7–29.6 ng mL^−1^), IAA (0.37–5.27 µg mL^−1^), and ABA (11.7–54.0 ng mL^−1^). The highest level of GAS_3_ was found for the strain Cap 03D *Arthrobacter* sp.—29.6 ng mL^−1^. The strains App 11D *Paenibacillus* sp. and Cap 07D *Bacillus* sp. produced the maximum amount of abscisic acid (54.0 and 37.0 ng mL^−1^), accordingly. The strains Cap 07 and Cap 286 *Bacillus* spp. were the most active producers of IAA, which is consistent with the results presented in Figure 3.

### 2.7. Plant Growth Promotion Activity of PGPB

The growth-stimulating activity was tested after growing wheat cv. Leningradskaya 6 for 4 days (Figure 4). All strains have shown the growth-stimulating activity increasing the root growth (by 8.3–19.4%) and shoot growth (by 0.6–8.4%) as compared to the control variant. The highest root growth-stimulating activity was detected in the Cap 07D, Cap 09D, and Cap 286 *Bacillus* spp. strains.

### 2.8. Hydroponic Experiments

We studied the effect of PGPB strains on wheat growth under osmotic stress. In order to create osmotic stress, we used PEG6000 because it is widely used in similar studies [34,44,48,49]. Previously, we tested the effect of PEG6000 on the studied strains. It was found that the addition of 12% PEG6000 to the LB medium during cultivation does not significantly affect the growth of the strains (Appendix A).

#### 2.8.1. Effect of PGPB on Wheat Morphological Characteristics Under Normal Conditions and Osmotic Stress

In the hydroponic experiments, the morphological characteristics of wheat plants under osmotic stress (generated by the addition of 12% PEG6000) were studied (Table 5, Figure 5).

The inoculation with all PGPB studied under control conditions (without osmotic stress) significantly increased the wheat root length by 9.3–15.9%. The root fresh weight (FW) also increased significantly (by 9.6–15.8%) in all control variants when inoculated with the studied bacteria.

In general, the increase in the root FW of Leningradskaya 6 was associated with intensive root formation after the inoculation with PGPB (Figure 5). This is also confirmed by the significant increase in the dry weight (DW) of roots (by 20.9–50.9%) after the inoculation with all PGPB studied in the control variants. The dry matter (DM) in roots is also increased by 4–35%. The strains Cap 03D *Arthrobacter sp.*, Cap 07D, and App 11D *Bacillus* spp. significantly increased the shoot length (by 3.9–5.2%) in control variants, whereas Cap 09 *Bacillus* sp. and Cap 286 *Paenibacillus* sp. had no significant effect on the shoot length.

Osmotic stress had a negative effect on wheat plants. The addition of 12% PEG6000 significantly decreased the length of roots by 12.1% and leaves by 15.3%. Moreover, the FW of the roots and shoots decreased by 50.6% and 39.1% (Table 5), and the DW of the roots and shoots also decreased (by 51.4% and 32.4%), respectively.

Inoculation with the studied PGPB significantly reduced the effect of osmotic stress. PGPB significantly increased the length of the roots and shoots (except for strain Cap 03D *Arthrobacter* sp., which unreliably increased the shoot length) in all variants with the addition of 12% PEG600. No differences in the effect of the studied strains on the wheat root length were detected (Table 5); the efficiency of PGPB was 9–12.1%. PGPB under osmotic stress also increased the shoot length (by 3.8–11.9%) and root length (by 9.4–12.5%). The strain App 11D *Paenibacillus* sp. Showed the highest efficiency. PGPB also increased the DW of the roots (by 11.2–75.8%) and shoots (by 12.5–47.0%) under osmotic stress, when the strains Cap 03D *Arthrobacter* sp. And App 11D *Paenibacillus* sp. Showed the highest efficiency. The increase in the DW of wheat plants upon PGPB inoculation was accompanied by an increase in dry matter (%) by 0.6–2.2% in roots and by 0.6–3.4% shoots (Figure 6).

#### 2.8.2. Content of Photosynthetic Pigments in Wheat Leaves

Inoculation with Cap 07D, Cap 09D, and Cap 286 *Bacillus* spp. Led to a significant increase in chlorophyll a content in wheat leaves in the control variants, whereas the strains Cap 03D *Arthrobacter* sp. And App 11D *Paenibacillus* sp. Had no effect on the chlorophyll a content (Figure 7a). The content of chlorophyll b did not increase significantly and the strains Cap 09D *Bacillus* sp. And App 11D *Paenibacillus* sp. Saw a decrease in chlorophyll b content compared to the control variant (without the addition of 12% PEG6000) (Figure 7b).

Osmotic stress significantly decreased the content of total chlorophyll by 17.5% (mainly due to a decrease in chlorophyll a) (Figure 8a) and the content of carotenoids by 14% in wheat leaves (Figure 8b).

The differences of the bacteria action on the photosynthetic pigments of the plants both in the control conditions and under osmotic stress were revealed. Bacteria under control conditions increased the content of total chlorophyll in wheat leaves by 5.5–13.4. The strain App 11D *Paenibacillus* sp. did not show its efficiency on the content of total chlorophyll and carotenoids in wheat leaves. The strain Cap 286 *Bacillus* sp. showed the highest efficiency in control conditions on the number of photosynthetic pigments: total chlorophyll and carotenoids. Inoculation with this strain under osmotic stress led to an increase in the content of chlorophyll and carotenoids to the level of the control variant.

#### 2.8.3. Effect of PGPB on Enzyme Activities of Wheat cv. Leningradskaya 6

The inoculation with PGPB of wheat plants modulated the peroxidase (POX) and catalase (CAT) activities (Table 6).

The peroxidase (POX) activity in the shoots of wheat var. Leningradskaya 6 in the control variant was 1.9 times lower than in the roots. The inoculation with Cap 03D *Arthrobacter* sp. and Cap 07D *Bacillus* sp. under control conditions (without PEG 6000) increased the POX activity in wheat roots by 25.8 and 16.7%, respectively. The inoculation with other studied strains under control conditions had no significant effect on the POX activity in the roots of wheat plants.

Under the conditions of osmotic stress, the POX activity in plant roots increased by 42.2%. The inoculation with bacteria under stress conditions decreased the POX activity in wheat roots (by 7.6–29%). When inoculated with bacteria under osmotic stress, the POX activity in shoots also decreased, except for strain Cap 286 *Bacillus* sp., which did not affect the POX activity in the shoots, and strain Cap 07D *Bacillus* sp., which, on the contrary, increased the POX activity in the shoots by 1.4 times.

The shoots of wheat plants have shown a higher catalase (CAT) activity, which was 2.57 times higher than its activity in roots. Inoculation with PGPB under control conditions resulted in a decrease in CAT activity in the roots (by 6.8–16.7% and shoots (by 3.1–19.8%) of wheat plants, respectively. Osmotic stress significantly decreased the CAT activity in the roots and shoots by 26.3% and 13%, respectively. The inoculation with PGPB under osmotic stress led to an increase in CAT activity in the roots (by 20.8–34.0%) and, on the contrary, a decrease in CAT activity in the shoots (by 0.9–9.7%), except for strain Cap 286 *Bacillus* sp., which did not significantly change the CAT activity in shoots.

#### 2.8.4. Effect of PGPB on MDA Content

One of the metabolites of lipid peroxidation is malondialdehyde (MDA), an increase in which shows that plants are subjected to oxidative stress. The inoculation with PGPB under control conditions did not affect the change in MDA in wheat roots, but decreased its content in the shoots. When 12% PEG 6000 was added, the MDA content increased significantly in both the roots (by 13%) and shoots (by 37%) (Figure 9a). This indicates that wheat shoots were more susceptible to osmotic stress. The strains studied had a protective effect on wheat plants. The content of MDA decreased significantly in roots (by 11–13%) and shoots (by 28–33%) when inoculated with PGPB under osmotic stress. This indicates the protective function of PGPB. The strains Cap 07D and Cap286 *Bacillus* spp. had the greatest effect on reducing the amount of MDA.

#### 2.8.5. Effects of PGPB on Endogenous Wheat Hormone Contents

We analyzed the levels of phytohormones IAA, GAS_3_, and ABA in the roots of plants inoculated with PGPB strains both under normal conditions and under osmotic stress. The influences of PGPB on the plant hormone contents in roots are shown in Figure 10.

In the control variant (uninoculated), the concentrations of IAA and GAS_3_ in wheat roots decreased under osmotic stress by 10.5% and by 11%, accordingly (Figure 10a,b). The ABA concentration in the control variant, on the contrary, increased by 48.4% under osmotic stress (compared to the uninoculated control variant) (Figure 10c).

The inoculation with PGPB mainly increased the contents of IAA, GAS_3_, and ABA compared to the uninoculated variant. However, the inoculation with strain Cap 07D *Bacillus* sp. did not change the GAS_3_ content in both variants and the inoculation with Cap 286 *Bacillus* sp. decreased the GAS_3_ in wheat roots under osmotic stress. The highest level of GAS_3_ was found in the roots of wheat plants inoculated with strains Cap 09D *Bacillus* sp. and App 11D *Paenibacillus* sp.

Under osmotic stress, the concentrations of IAA and GAS3 decreased when bacteria were inoculated, while the concentration of ABA, on the contrary, increased by 25–60.2%. At the same time, under the conditions of osmotic stress, strains Cap 03D *Arthrobacter* sp. and Cap 07D *Bacillus* sp. did not change the ABA content compared to the control without inoculation, and the strain Cap 286 *Bacillus* sp. significantly reduced the ABA content in wheat roots.

## 3. Discussion

Drought is a serious problem, limiting the production of crops, including cereals. PGPGs are known to help wheat plants overcome drought stress [50,51,52,53]. The wheat root system is severely depressed in times of drought stress, including the root biomass, surface area, and length. Moreover, the level of photosynthetic pigments and wheat yield decreased under drought stress [52,54]. The inoculation with PGPB of wheat plants under drought conditions helps to overcome plant stress, because bacterial phytohormones improve the plant growth, plasma membrane integrity, and photosynthesis of plants [34,55].

The bacteria selected for our experiments preliminarily showed high growth-stimulating activity on wheat plants and were identified as *Bacillus* spp. (strains Cap 07D, Cap 09 D and App 11D), *Paenibacillus* sp. (strain Cap 286), and *Arthrobacter* sp. (strain Cap 03D). All the PGPB studied (with exception Cap 03D) produced exopolysaccharides, which protect plants’ roots from desiccation [56]. The strains *Bacillus* spp. Cap 09D and App 11D were able to grow at 50 °C and 10% NaCl, which allows them to be used not only under drought conditions but also under high soil salinity.

All the strains were found capable of producing IAA, ABA, and GAS_3_. It is known that PGPB from these genera produce various phytohormones, such as IAA, ABA, GA, and CK [57,58,59,60]. Indeed, all our strains studied produced total indoles (including IAA), according to the Salkowskiy test, but the most effective producers were Cap 07D and Cap 286. Indeed, most PGPB produced auxin, which is a major component of the bacterial promotion of plant growth and development [61]. There are several IAA biosynthetic pathways in PGPB and these pathways can intersect with each other [62]. IAA promotes plant growth, the differentiation of vascular tissues, the stimulation of cell division [63], and bacterial colonization of plant roots [64,65]. IAA is involved in the regulation of the adaptive response of plants to drought stress. Auxin acts on gene expression through a family of functionally distinct transcription factors, the DNA-binding auxin response factors (ARFs) [66]. Different ARFs regulate soluble sugars, promote root development, and maintain the chlorophyll content under drought and salt stress conditions, helping plants to adapt to these stresses [66]; IAA is one of the most studied hormones in various plant crops, including grain cultures [61].

It was shown that IAA-producing bacteria reduced osmotic stress in different plants [67,68,69]. In our experiments, drought stress had a negative effect on the wheat length and weight. Inoculation with PGPB increased the root growth and lateral root formation. It is known that IAA-producing bacteria control endogenous IAA levels in plant roots by regulating auxin-responsive genes, which changes the rice root architecture [70]. Moreover, auxin-producing bacteria enhance both the root and shoot growth [71]. IAA biosynthesis by PGPB helps in successful root colonization, which leads to a decrease in the drought stress in different plants [72]. Our results are in agreement with the previously obtained data [34,73]. It was found that the bacteria *Bacillus amyloliquefaciens* 5113 and *Azospirillum brasilense* NO40 also influenced the wheat root system and improved the drought stress tolerance in wheat [74].

Along with auxin synthesis, all the studied bacteria synthesize gibberelic (GAS_3_) and abscisic acids (ABA), which are hormone antagonists in various physiological processes [75]. ABA is the plant hormone which is synthesized in response to abiotic stresses and activates the genes responsible for stress resistance [76]. Under drought stress, ABA affects different aspects of plant growth, such as reducing photosynthesis, inhibiting leaf surface area expansion, decreasing lateral root initiation [77], promoting stomatal closure [76], increasing hydraulic conductivity, stimulating root cell elongation [78], and promoting plant resistance to drought [79]. It is known that inoculation with ABA-producing bacteria against drought conditions can lead to an increase in ABA content in plants, which become more resistant to stress [35,80,81,82].

Gibberellic acids (GASs) modulate various developmental processes in plants, including seed germination, stem elongation, flowering, ripening [83], the chlorophyll content, and the photosynthetic activity [84]. It is known that the exogenous application of gibberellic acid mitigates drought stress in wheat plants [85]. It was shown that GAS-producing PGPB *Leifsonia soli* SE134 and *Enterococcus faecium* LKE12 stimulated shoot growth in mutant rice plants (deficient in gibberellin synthesis), compensating for the absence of plant gibberellins [86,87]. Inoculation with GAS-producing PGPB decreased drought stress [88,89,90].

Under drought conditions, the CO_2_ influx is restricted by limiting the stomatal and mesophyll conductance [91], resulting in reduced carbon assimilation by the photosynthetic apparatus [12,81]. It was shown that, under drought stress in wheat plants, ATP synthesis was impaired due to decreased electron transport and membrane damage, and the content and efficiency of the key photosynthetic enzyme Rubisco (Ribulose bisphosphate carboxylase/oxygenase) were reduced [92]. Bacterial inoculation under drought stress led to an increase in photosynthetic pigments [93] and maximized the efficiency of photosystem II by improving the stomatal conductance, CO_2_ assimilation, and transpiration rate [81]. In our experiments, osmotic stress significantly decreased the content of total chlorophyll by 17.5% and the content of carotenoids by 14% in wheat leaves in our experiment. PGPB under osmotic stress increased the total chlorophyll content by increasing chlorophyll *a*. It is likely that this favorable effect on the chlorophyll content of the PGPB was due to the production of GAS, which regulates the chlorophyll content. The strain Cap 286 *Paenibacillus* sp., in addition to synthesizing GAS_3_, has nitrogen-fixing activity and a maximally increased chlorophyll content in wheat plants. The strong linear relationships between nitrogen and both ribulose 1,5-bisphosphate (RuBP) carboxylase and chlorophyll were established early [94]. Therefore, the N-fixing bacterium *Paenibacillus* sp. Cap 286 increased the chlorophyll content in wheat leaves due to the nitrogen accumulation in plants.

Catalase (CAT) and peroxidase (POX) are known antioxidant enzymes, whose activity leads to a decrease in the production of highly reactive hydroxide ions. Peroxidases promote the conversion of hydrogen peroxide (H_2_O_2_) to water. H_2_O_2_ is a main component of stress response regulation in wheat [95] and other cereals [96,97]. Both antioxidant enzymes CAT and POX take part in counteracting the oxidative damage in wheat during drought stress [73,98,99,100]. In some studies, the POX activity increased under drought conditions indicating the adverse effects on the morphological and physiological characters in several plants [101,102,103,104]. The increase in the POX activity is indicative of an excessive formation of ROS during photorespiration and photosynthesis in peroxisomes and chloroplasts [105]. The POX activity in the wheat roots and shoots in our experiments increased under osmotic stress, which is consistent with earlier results obtained in wheat cultivation [74,104]. However, there are contrary results indicating a decrease in wheat the POX activity under drought stress [106,107]. Perhaps the contradictions in the results are explained by the different wheat genotypes used in the experiments and the different durations of stress.

In our experiments, PGPB modulate the POX and CAT activities after 7 days of osmotic stress influence. The POX activity in the roots and shoots of wheat plants mainly decreased after the inoculation with PGPB under drought conditions (with the exception of Cap 07D *Bacillus* sp.). Similar results were received early [74]. The authors attributed the decrease in peroxidase activity to the protective effect of the bacteria. In general, PGPB under osmotic stress conditions resulted in an increase in CAT activity in the roots and, conversely, a decrease in CAT in the shoots. Bacterial inoculation under drought has different effects on the CAT activity in different plants, both increasing and decreasing the activity of the enzyme [41,108,109,110]. Our studies are in accordance with the results obtained early [44], where it was reported that the inoculation of wheat plants with PGPB (*Bacillus* sp. D13) under drought stress decreased the CAT activity. At the same time, there is an opposite result regarding the increase in CAT activity in wheat plants when inoculated with the bacteria *Bacillus subtilis* under drought conditions [41]. These differences in the CAT response to the inoculation of wheat with *Bacillus* sp. under drought stress may be explained by the genetic differences in wheat plants and PGPB.

In our experiments, osmotic stress leads to an increase in MDA content in wheat roots and shoots. It was previously shown that the MDA content in various plants such as *Brassica napus* [111], *Medicago sativa* [112], *Zea mays* [113], and *Triticum aestivum* [34,114] increased under drought stress. In general, the MDA content (a product of lipid peroxidation) is a reliable indicator of oxidative membrane damage due to oxidative stress [115]. However, there are studies that did not reveal a correlation between the MDA content in wheat and wheat drought tolerance [116,117,118]. This appears to indicate the complex mechanisms of drought tolerance in wheat that may vary among genotypes. Bacterial inoculation significantly reduced the MDA content in the roots and shoots under drought stress conditions in our experiments. Of all the bacteria studied, the most effective were the strains Cap 286 and Cap 07D *Bacillus* spp. Other studies have found that the PGPB significantly decreased the MDA content under drought conditions (compared to non-inoculated plants) and protected plants against stress [34,119]. It follows that PGPB prevent the damage of plant cell membranes and it is one of the key points for a plant to resist drought stress in the plant–bacterial association.

The PGPB studied also modulated the wheat hormonal system, which plays a pivotal role in regulating the plant stress resistance [120]. PGPB mitigate the adverse effects of osmotic stress by increasing the ABA levels in wheat roots by 25–60.2%. Our results on the increase in ABA in wheat plant roots when plants are inoculated with ABA-producing bacteria under osmotic stress conditions are consistent with the previous results, obtained for other plants [80,81]. However, information on the effect of PGPB on the ABA levels in plants is limited and contradictory [121], as both increases and decreases in ABA levels have been reported [80,81,122]. These results suggest a complex hormonal regulation of plant drought stress processes by bacteria.

In general, PGPB isolated from drought-tolerant plants and producing ABA well protected wheat plants under osmotic stress. Bacterial ABA, which is a key phytohormone in the plant response to drought stress, probably influenced the change in the plant hormone background and contributed to the protection of wheat plants. However, the bacteria also produced other phytohormones (IAA and GAS_3_) and had other beneficial properties such as the production of exopolysaccharides and siderophores, which also provided the resistance of wheat to osmotic stress. It should be noted that strains Cap 09D *Bacillus* sp. and App 11D *Paenibacillus* sp. were capable of growing at temperatures of 50 °C and under saline conditions, which is very important for the selection of bioproducts for use in arid climate conditions, since high temperatures and salinity are often found together in natural arid conditions. Furthermore, we are planning to test the efficiency of PGPB strains in the pot and field experiments on wheat cultivation in arid regions.

Recently, a number of successful experiments have been conducted on the use of PGP bacteria of different species in wheat cultivation, both in model experiments under osmotic stress [34,44,48] and in vegetation and field experiments [28,49,52,123]. Effective inoculation with *Bacillus rugosus* MMH101under drought stress resulted in an increase in the shoot length (108.5%), and root length (134.9%) of wheat cultivar Gimmeza-9 [49]. Rashid et al. [28] studied the effect of *Bacillus megaterium* and *B. licheniformis* on wheat plants under drought conditions. The bacteria possessed ACC deaminase activities and IAA production. In response to drought stress, *B. megaterium* induced three new polypeptides, indicating proteomic changes. Bacterial inoculation increased photosynthetic pigments and osmolytes in wheat plants. Moreover, bacteria modulated the expression of genes involved in hormone signaling and the stress response. By analyzing the literature, we can conclude that the PGPB-mediated drought tolerance in wheat involves complex interactions between microbes, hormones, and proteomic dynamics.

Currently, there is a huge need for robust and environmentally friendly farming strategies under biotic and abiotic stresses. Products aimed to protect agricultural crops from stresses such as drought, waterlogging, salinization, heavy metals, and pathogenicity will become increasingly important in the future due to the scenario of global climate change. In this regard, the use of PGPB could be a means of protecting plants from stress and lead to promising solutions for sustainable and environmentally friendly agriculture. There is a huge growing market worldwide for microbiological preparations to improve plant growth and increase yields under biotic and abiotic stress conditions with an annual growth rate of about 10% [124]. However, researchers must provide to the market the strains of PGPB which should be resistant to the extreme conditions of temperature and humidity, be susceptible to applied agrochemicals and pesticides, and demonstrate a high colonization and competition ability with the resident microorganisms.

## 4. Materials and Methods

### 4.1. Plant Samples for Bacterial Isolation

Plants and seeds of *C. album* were collected in Izobilny district of Stavropol Territory (near the town of Izobilny, N_45°23′36″, E_41°45′52″). Plants and seeds of *A. pseudoalhagi* were collected in Neftekumskiy district of Stavropol Territory (near the village Achikulak, N_44°33′43″, E_44°49′59″). Samples were taken at five points in each place. Plants were taken together with roots and adjacent soil. Each plant was taken with root and placed into craft bag, then transferred to the laboratory.

### 4.2. Isolation and Screening of PGPB for Hydroponic Experiments

PGPB were initially isolated from plant seeds, leaves, and stems of *A. pseudoalhagi* and *C. album* according to the protocol described by Pishchik et al. [58].

Plant material (10 g of seeds or aboveground parts) was placed to flasks (with 100 mL sterile water). The flasks were set on the ultrasonic bath (Bandelin; 50 Hz) for 10 min. Then, 0.1 mL of various dilutions was inoculated on Petri dishes containing LB (Luria Bertani, Sigma-Aldrich, St. Louis, USA) agar medium and Ashby medium (g L^−1^: sucrose—20.0, K_2_HP0_4_—0.2, MgSO_4_·7H_2_O—0.2, NaCl—0.2, K_2_SO_4_—0.2, CaCO_3_—5.0, and agar—18.0). Bacteria were grown for 3–5 days. Then, the strains were isolated and screened for PGP activity via the method described earlier [58]. The seeds of cv. Leningradskaya 6 (30 seeds per variant) were dipped for 30 min into bacterial cultures (1–3 × 10^5^ cells per mL) or sterile water, and were used as a control. Then, the seeds were replaced in sterile Petri dishes (in triplicate) with filter paper and 10 mL of sterile water. Grain germination was taken into account on the 4th day of the incubation at 22 °C. Wheat seedlings were measured and weighted.

The strains which showed the greatest plant growth promotion activity were selected for the hydroponic experiments. To study the effect of 12% PEG 6000 on bacterial growth, we grew the bacteria on LB medium and on LB medium supplemented with 12% PEG 6000 for 72 h at 28 °C at 180 rpm. Then, we measured the absorbances of the bacteria on a Biomate 160 spectrophotometer (Thermo Scientific™, Madison, WI, USA) at 600 nm.

### 4.3. Bacterial Identification

Total DNA from bacterial strains was isolated with use of Monarch^®^ Genomic DNA Purification Kit (New England BioLabs^®^ Inc., Ipswich, MA, USA) according to the producer’s instructions. The universal primers 27f (50-AGA GTT TGATCM TGG CTC AG-30) and 1525r (50-AAG GAG GTG WTC CAR CC-30) [125] were used for amplification of DNA fragments in C1000 TouchTM Thermal Cycler (Bio-Rad, Hercules, CA, USA). PCR was performed in 30 µL reaction mixtures containing 10× Taq Turbo buffer (Evrogen, Moscow, Russia), 150 µM dNTPs (Evrogen, Moscow, Russia), 5 pM of each primer, and 2 U of Taq polymerase (Evrogen, Moscow, Russia). The initial DNA denaturation was performed at 95 °C for 3 min, then followed by 30 cycles of denaturation at 95 °C for 30 s; the primer annealing at 54 °C for 30 s; the elongation at 72 °C for 30 s; and final elongation at 72 °C for 5 min. The obtained PCR fragments were isolated from 1% agarose gel using Monarch^®^ DNA Gel Extraction Kit (New England BioLabs^®^ Inc., Ipswich, MA, USA) and sequenced using an ABI PRISM 3500xl capillary electrophoresis sequencing station (Applied Biosystems, Waltham, MA, USA) according to the manufacturer’s protocol.

The strains were identified by comparing the obtained nucleotide sequences of the 16S rRNA gene with the RDP [126] and GenBank [127] databases. The 16S rRNA gene sequences of all the isolates were deposited in the GenBank database with the accession numbers PQ516836–PQ516840. Multiple sequence alignment was conducted by using the Clustal X 1.8 software package. Phylogenetic trees were generated by the Neighbor-Joining method using the MEGA 11 software package, and confidence was tested by bootstrap analysis with 500 repeats.

### 4.4. Nitrogen-Fixing Ability

NFA was assayed on the bacterial growth on nitrogen-free Ashby medium (g L^−1^: sucrose—20.0, K_2_HP0_4_—0.2, MgSO_4_·7H_2_O—0.2, NaCl—0.2, K_2_SO_4_—0.2, CaCO_3_—5.0, and agar—18.0). The bacteria were incubated at a temperature of 28 °C for 48 h.

### 4.5. Indole Production Activity

Indole production activity was studied using Salkowski reagent [128]. The studied bacteria were grown in 100 mL R2A medium (yeast extract 0.5; proteose peptone 0.5; casein hydrolysate 0.5; glucose 0.5; starch soluble 0.5; sodium pyruvate 0.3; dipotassium hydrogen phosphate 0.3; and magnesium sulfate 0.024) with 500 mg L^−1^ tryptophan for 72 h at 28 °C at 180 rpm. Then, bacterial cells were centrifuged twice during 90 s at 13,500 on Eppendorf AG 5804 R (Hamburg, Germany) for 15 min at 5000 rpm, and the supernatant fluid was subsequently filtered through 0.22 µm. Then, 1 mL Salkowski reagent (1 mL 0.5 M FeCl_3_ in 50 mL of 35% HClO_4_) was added to 0.5 mL of supernatant. After 30 min of storage in the dark, the absorbances of colored supernatants were measured on a spectrophotometer Biomate 160 (Thermo Scientific™, Madison, USA) at 540 nm. The indole contents were calculated used IAA standard curve based on dilution series of authentic IAA (0.5; 1.0; 5.0; and 50.0 µg/mL).

### 4.6. Phosphorus Solubilization

The ability of isolated endophytic bacteria to dissolve calcium orthophosphates was assayed on Muromtsev (g L^−1^: glucose—10; asparagine—1; K_2_SO_4_—0.2; MgSO_4_x7H_2_O—0.2; agar—20; yeast extract—0.02; and tap water, pH 6.8) as described by Chebotar et al. [129]. Calcium phosphates Ca_3_(PO_4_)_2_ were introduced into the nutrient medium by precipitation. To do this, 3.4 g of CaCl_2_ and 3.8 g Na_3_PO_4_ were added to the sterile molten agar medium (based on one liter) as they dissolved. Then, 1.5 g of freshly formed Ca_3_(PO_4_)_2_ was formed per one liter of medium. The nutrient medium was poured into Petri dishes and tested strains were spread on Petri dishes. The Petri dishes were incubated at a temperature of 28 °C for 120–216 h. The phosphorus-solubilizing activity was judged as the appearance of clear zones around the growth area of a bacterial sample spotted on the plate.

### 4.7. Production of Exopolysaccharides

Bacterial exopolysaccharides were assayed as described by Shaffique et al. [27] with the modification. The Congo red agar assay was used to identify the presence of exopolysaccharides if produced by the bacterial isolate. Assay plates were prepared by mixing in LB broth (25 g L^−1^), agar (20 g L^−1^), sucrose (10 g L^−1^), and Congo red (0.8 g L^−1^). All chemicals were well-mixed and then autoclaved. Petri dishes with Congo red agar were inoculated with the strains studied. PGPR grow for 5 days at 30 °C. The black colonies are visible against the red background, indicating the presence of polysaccharides.

### 4.8. Bacterial Phytohormone Assay

The strains studied were grown on liquid R2A medium for 72 h at 28 °C at 180 rpm. The control was uninoculated R2A medium. The concentrations of bacterial phytohormones—indole acetic acid (IAA), abscisic acid (ABA), and gibberellic acid (GAS_3_) were determined using the high-performance liquid chromatograph VARIAN 212 LC with a mass-selective detector (Varian 500 MS system) as described early [58]. Detection of IAA, ABA, and GAS_3_ was carried out using ESI- (electrospray) ion at 174, 265, and 345 *m*/*z*, respectively. For phytohormones determination, 50 mL of liquid culture medium (and 50 mL of sterile liquid medium used as control) was taken and centrifuged at 5000 rpm for 5 min. The supernatant was poured into a separating funnel. The combined supernatant in the separating funnel was acidified with 10% acetic acid solution to pH 2, after which phytohormones were extracted three times with 10 mL of ethyl acetate. The upper ethyl acetate layer was drained through anhydrous sodium sulfate and evaporated to dry residue on a rotary vacuum evaporator at a temperature not exceeding 40°C. The residue in the distillation flask was washed with 2 mL of mobile phase A. Chromatography was carried out in gradient mode (phase A, methanol + 0.1% formic acid; and phase B, deionized water +0.1% formic acid). A Cosmosil C18 4.6 ID × 150 mm column was used in the chromatographic system. The chromatograph was calibrated using SIGMA-ALDRICH internal standards for pure phytohormones. The identification of hormones was carried out in mass–mass mode.

### 4.9. Hydroponic Experiments

Seeds of wheat plant cv. Leningradskaya 6 were obtained from Leningrad Scientific Research Institute of Agricultural Science “Belogorka” (St. Petersburg, Leningrad Region, Russia). The wheat seeds were sterilized in 70% ethyl alcohol for 60 s, washed with tap water, and then placed into 2% sodium hypochlorite solution for 15 min and rinsed in sterile water 7 times. After that, the seeds were placed in Petri dishes in the thermostat at 25 °C for 48 h for the germination. The germinated seeds (three-day-old seedlings) were placed between two layers of hydrophilic tissue at a distance of 2 cm from each other. The tissue was rolled up and placed in 300 mL vessels with Knop nutrient solution (containing CaNO_3_—1 g, KH_2_PO_4_—0.25 g, MgSO_4_—0.25 g, KCl—0.125 g, and FeCl_3_—0.0125 g per 1 L). Bacterial inoculation was carried out immediately after placing the seedlings in Knop’s solution. PGPB strains Cap 03D, Cap 07D, Cap 09D, App 11D, and Cap 286 were used as inoculants. Bacteria were added to the Knop solution in concentrations of 3 × 10^5^ CFU in mL. Bacteria were precultured on liquid medium; their titer was determined and diluted to the required concentration.

Osmotic stress was created by addition of polyethylene glycol (PEG 6000) at a concentration of 12% on the 7th day of experiment. PEG concentration was chosen in our preliminary experiments (see Appendix A), taking into account the recommendations of other researchers [34]. The vessels with wheat seedlings were placed on an installation for vegetation experiments. The vegetation was maintained in a 16 h photoperiod (16 h of light and 8 h of darkness) under a constant temperature of 22 °C during the day and 18 °C at night, with a light intensity of 23–25 klx per m^2^. The vegetation period lasted for 14 days (7 days with osmotic stress exposure and inoculation with PGPB (10 days of exposure)). After the end of experiments, the 14-day-old seedlings were taken to determine the contents of photosynthetic pigments and plant ferments and the levels of hormones. Seedlings were measured and weighted. To determine dry weight (DW), plants were dried for 5 h at 70 °C, and then for 30 min at 105 °C. Dry matter (DM) was calculated as % of plant DW in plant fresh weight FW.

### 4.10. Analysis of Plant Phytohormones

Plants phytohormones (IAA, ABA, and GAS_3_) were determined as described early [59]. First, 10 g of leaves were homogenized with 80% methanol at 4 °C and evaporated. One half of remaining aqueous phase was acidified with 10% solution of muriatic acid to pH 2.5–3 and extracted three times in a separating funnel with 30 mL diethyl ether (phase A). The final extracts were evaporated to a dry residue and dissolved in 2 mL of mobile phase A. Detection of IAA, ABA, and GAS_3_ was carried out using ESI- (electrospray) ion at 174, 265, and 345 *m*/*z*, respectively. Chromatography was performed in gradient mode.

Cosmosil C18 4.6 ID × 150 mm column was used in the chromatographic system. The liquid chromatograph VARIAN 212 LC with mass selective detector (Varian 500 MS system) was calibrated using Sigma-Aldrich internal standards for pure hormones. The hormones were identified in MS/MS mode.

### 4.11. Chlorophyll and Carotenoids Analysis

For the determination of chlorophyll a (Chl a), chlorophyll b (Chl b), and carotenoids (Car), concentrations of 0.2 g of fresh leaves were ground in a porcelain mortar with a small amount of acetone and sand. The ground mass was placed in tubes containing 10 mL of 80% acetone for 24 h at 20 °C. The tubes were centrifuged at 20,000× *g* for 20 min. Then, supernatant was transferred into a 50 mL volumetric flask and made up to the mark with acetone. The absorbance of the resulting supernatant was measured at 663, 645, and 470 nm using a spectrophotometer (ModelSpekol 1500 (Jena, Germany). The contents of chlorophyll a, chlorophyll b, and total carotenoid were calculated after [130]. The pigments concentrations in mg/dm^3^ were calculated using the following formulae:C chl a (mg/dm^3^) = (9.784 × OD662) − (0.99 × OD644)
C chl b (mg/dm^3^) = (21.426 × OD644) − (4.65 × OD662)
C car (mg/dm^3^) = (4.695 × OD440.5) − (0.268 (C chl + C chl b))
where: OD—absorbtion value at the corresponding wawelenghth 

To recalculate the pigments concentrations X in mg/g, the following formula was used:X = C × V/n × 1000
where C—pigment content (mg/dm^3^), V—extract volume in cm^3^, and n—sample weight in grams.

### 4.12. Malondialdehyde (MDA) Content

Lipid peroxidation (LPO) occurs in response to oxidative stress, giving rise to unsaturated aldehydes as malondialdehyde (MDA). The malondialdehyde (MDA) content was measured using the method described by Uchiyama and Mihara [131]. First, 0.3 g of fresh leaves was homogenized and filtered. The reaction medium consists of 0.3 mL plant homogenate, 3 mL 1% H_3_PO_4_, and 1 mL 0.6% thiobarbituric acid (TBA) aqueous solution. The homogenate was put in a water bath at 95–100 °C for 60 min, after which the samples were cooled down. Then, 4 mL of n-butanol were added and centrifuged for 10 min at 10,000× *g*. The absorbance of butanol extract was measured at 532 nm and 600 nm using the spectrophotometer Model Spekol 1500, Jena, Germany. The concentration of TBA-reactive products was expressed in MDA µM g^−1^ FW.

### 4.13. Catalase (CAT)

The CAT (EC1.11.1.6) was measured using the method described by Aebi [132]. This method is based on the effect of CAT on hydrogen peroxide (H_2_O_2_) and the measurement of the ultraviolet absorption of H_2_O_2_ at 240 nm. The reaction mixture at a volume of 3 mL contained 0.1 M of sodium phosphate buffer (pH 7.0), 2 mM of H_2_O_2_, and 0.2 mL of enzyme extract. For the calculation of the activity, the extinction coefficient of 0.036 mM^−1^ cm^−1^ was employed. CAT activity was expressed in µM of H_2_O_2_ per mg^−1^ protein min^−1^.

### 4.14. POX Activity

Activity of POX in plants was determined as described by Ermakov et al. [133]. A 200–500 mg weight of plant material is finely ground in a porcelain mortar with acetate buffer with pH 5.4 and transferred into a 50 cm^3^ measuring flask. After 10 min of infusion, the extract is centrifuged at 4000 rpm. Two quartz cuvettes with a working length of 20 mm are filled with 2 cm^3^ of fermentation fume or centrifugate and 2 cm^3^ of benzidine. To prepare a solution of benzidine, 100 cm^3^ of distilled water is poured into a 200 cm^3^ measuring flask, and 2.3 cm^3^ of glacial acetic acid and 184 mg of benzidine are added. The flask is heated in a water bath at 60 °C, shaking constantly; after complete dissolution of the benzidine, 5.45 g sodium acetic acid is added to the flask, cooled, and brought to the mark with water. The cuvettes in the photoelectrocolorimeter are placed against the red filters (590 nm). The measurement is carried out using a red-light filter (at 590 nm) in the following sequence. At the beginning, with the left drum, set the galvanometer pointer to zero; then, with the right drum, move the galvanometer pointer to the leftmost position (D = 0.125 or 0.250). After that, 2 cm^3^ of water is poured into the control right cuvette, and 2 cm^3^ of 0.3% hydrogen peroxide from a wide-bore pipette is poured into the left experimental (measuring) cuvette. When the first drop of hydrogen peroxide is added, the stopwatch is started. The galvanometer hand starts to deviate from the edge of the scale to the zero division. The stopwatch is stopped when the galvanometer hand reaches zero. From the reaction rate found, the activity A of the enzyme is calculated as follows:
*A* = *D* × *a* × *b* × *c*/*f* × *t*

where:*D*—optical density, equal to 0.125 or 0.250;*a*—ratio of the amount of liquid taken for preparation of the extract to the mass of raw tissue, cm^3^/g;*b*—degree of additional dilution of the extract after centrifugation;*c*—degree of constant dilution of the extract in the reaction mixture in the cuvette;*f*—layer thickness (2 cm);t—time, s.

POX activity was expressed in units, each representing the absorbance value per second per g^−1^ FW.

### 4.15. Statistical Analysis

The independent experiments were performed to assess the wheat growth parameters. We carried these out three times with 15 seedlings of Leningradskaya 6 (*n* = 45). We also used three biological replicates, with 30 seedlings per replicate, to assess the biochemical parameters of plants (*n* = 9). The data were statistically evaluated using STATISTICA-11 and subjected to a one-way analysis of variance (ANOVA). The mean values are shown as error bars representing standard errors of the means in all the figures. The data are presented as average mean standard error (SEM). Duncan’s multiple-range test was performed to determine significant differences between individual means. The differences between the means were determined at the level of significance *p* < 0.95.

## 5. Conclusions

PGPB from the genera *Bacillus* spp., *Paenibacillus* sp. and *Arthrobacter* sp. isolated from arid plants *A. pseudoalhagi* and *C. album* stimulated growth and induced wheat (cv. Leningradskaya 6) drought tolerance. All bacteria produced IAA, GAS3, ABA, and siderophores. Most of them produced exopolysaccharides (with the exception of *Arthrobacter* sp. (Cap 03D). The strain Cap 07 *Bacillus* sp. possessed phosphate-mobilizing activity. The strain Cap 286 *Paenibacillus* sp. was able to fix nitrogen. Two strains from the genus *Bacillus* (Cap 09D and App 11D) grew at 50 °C and 10% NaCl. All PGPB significantly protected wheat plants under drought conditions in vegetative experiments. PGPB significantly increased the length and weight of wheat roots and shoots and decreased the content of MDA, indicating the protective effect of bacteria. PGPB modulated enzyme activities, mainly reducing the POX activity in wheat roots and shoots and differently affecting the CAT activity, increasing it in roots and decreasing it in shoots. PGPB increased the total chlorophyll content, predominantly increasing the chlorophyll *a* content in the leaves of wheat plants. PGPB modulated endogenous plant hormones, increasing the ABA content in wheat roots by 25–60.2% and, thus, helping plants overcome osmotic stress. All the bacteria studied in our experiments can be offered for further study in vegetative and field experiments on wheat cultivation under drought stress, and strains Cap 09D *Bacillus* sp. and App 11D *Paenibacillus* sp. also under soil salinization.

## Figures and Tables

**Figure 1 plants-13-03381-f001:**
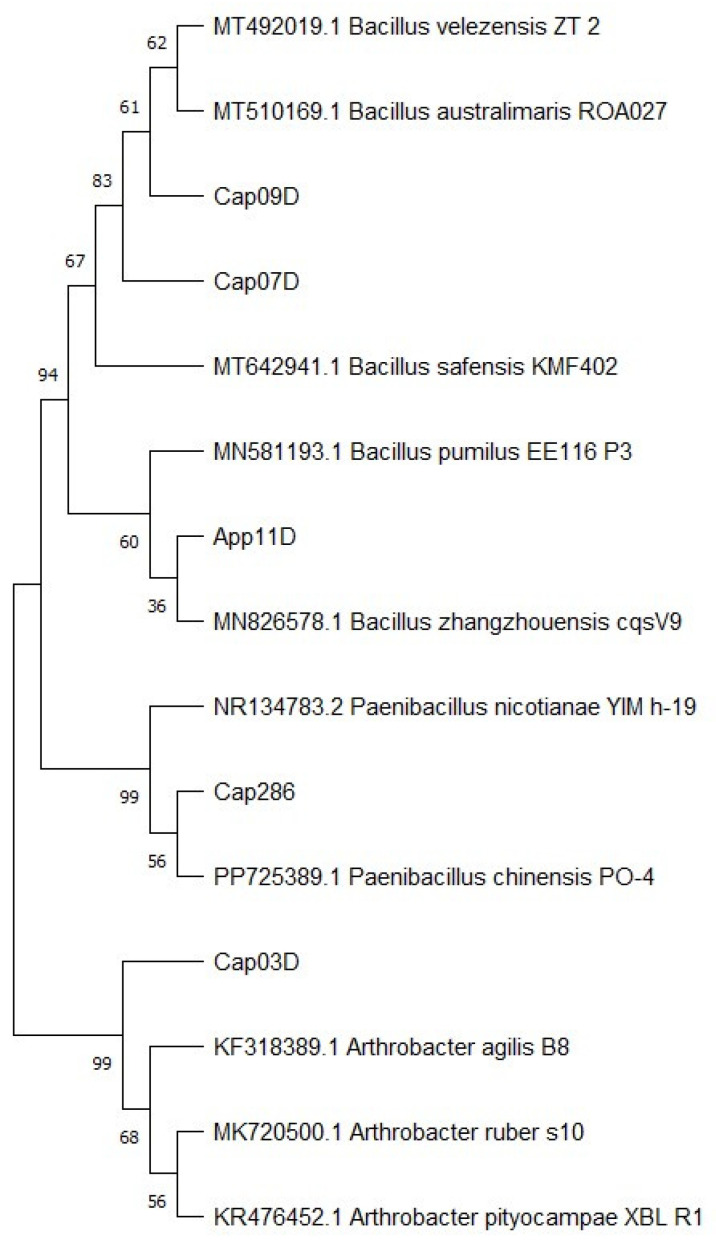
Phylogenetic analysis of 16S rRNA sequences of the epiphytic bacteria isolated from drought-tolerant plants. Notes: The Neighbor-Joining clustering method with bootstrap value (500 replicates), pairwise deletion algorithm, and p distance mathematical model were used.

**Figure 2 plants-13-03381-f002:**
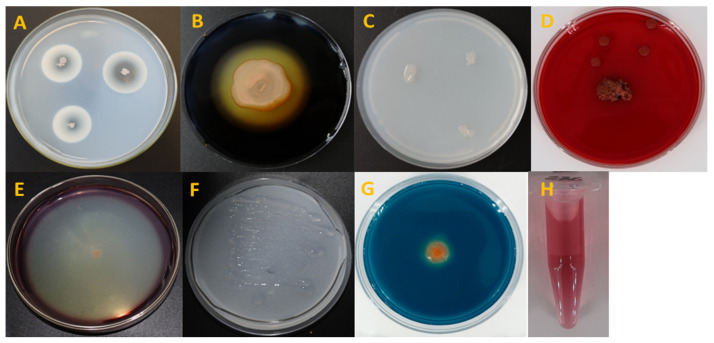
Metabolic activities of *Paenibacillus* sp. Cap 286. Notes: (**A**) Proteolytic activity; (**B**) amylolitic activity; (**C**) phosphate-mobilizing activity; (**D**) exopolysaccaride activity; (**E**) lypolitic activity; (**F**) nitrogen-fixing activity; (**G**) siderophore-producing activity; and (**H**): indole-producing activity.

**Figure 3 plants-13-03381-f003:**
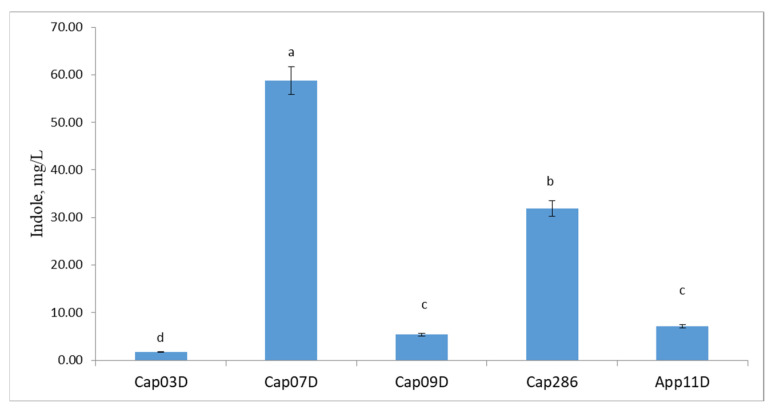
Indole production activities by PGPB studied with addition of tryptophan in the concentration 500 mg L^−1^. Notes: Cap03D: *Arthrobacter* sp. Cap 03D; Cap07D: *Bacillus* sp. Cap 07D; Cap09D: *Bacillus* sp. Cap 09D; Cap286: *Paenibacillus* sp. Cap 286; and App11D: *Bacillus* sp App 11D. Bars show ±SEM. Values in columns followed by different letters (a–d) are significantly different at *p* ≤ 0.05, as determined by Duncan’s multiple-range test.

**Figure 4 plants-13-03381-f004:**
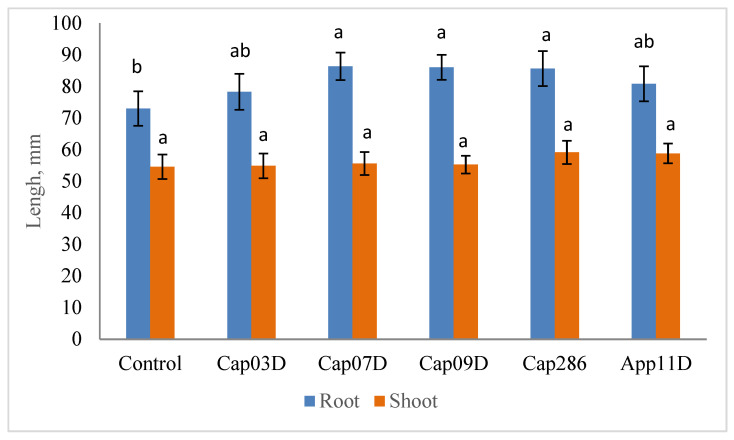
Effect of inoculation with PGPB on root and shoot lengths. Notes: The seedlings of wheat cv. Leningradskaya 6 were grown for 4 days. Control: non-inoculated wheat plants. Cap 03D, Cap 07D, Cap 09D, App 11D, and Cap 286: wheat plants inoculated with appropriate strains. Bars show ±SEM. Values in columns followed by different letters (a,b) are significantly different at *p* ≤ 0.05, as determined by Duncan’s multiple-range test.

**Figure 5 plants-13-03381-f005:**
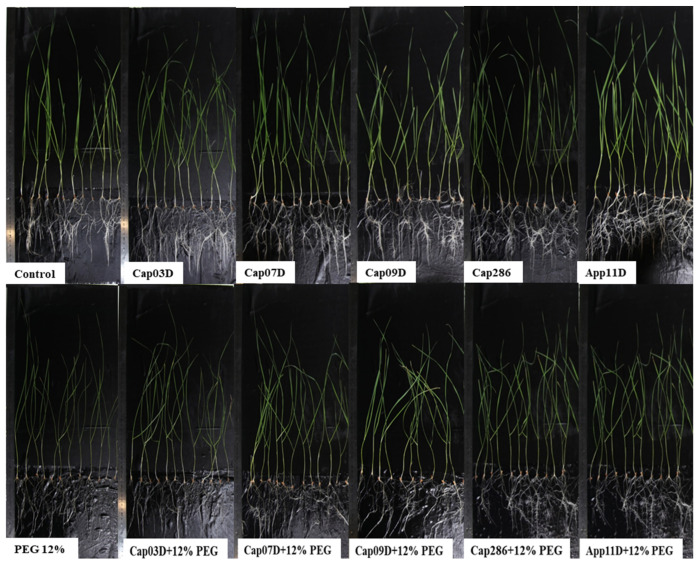
Effect of inoculation with PGPB on wheat plants cv. Leningradskaya 6 grown under normal and osmotic stress conditions. Notes: Control: non-inoculated wheat plants. PEG 12%: the addition of 12% PEG 6000; Cap03D, Cap07D, Cap09D, App11D, and Cap286: wheat plants inoculated with appropriate strains. Wheat plants cv. Leningradskaya 6 were grown under hydroponic conditions for 14 days (7 days of exposure to osmotic stress).

**Figure 6 plants-13-03381-f006:**
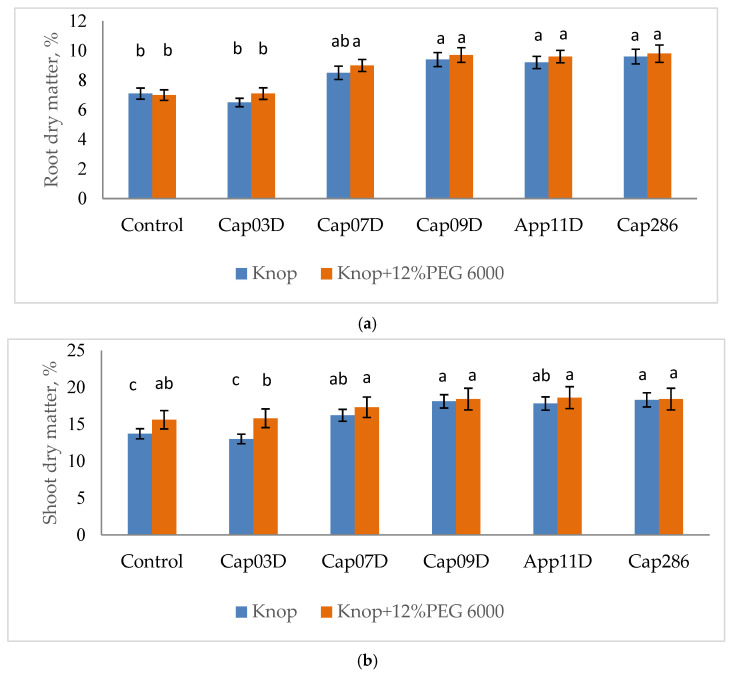
Effect of inoculation with PGPB on dry matter content in roots (**a**) and shoots (**b**) of wheat cv. Leningradskaya 6, grown under normal and osmotic stress conditions. Notes: Control: non-inoculated wheat plants, grown under control conditions (Knop) and under osmotic stress (the addition of 12% PEG 6000). Cap 03D, Cap 07D, Cap 09D, App 11D, and Cap 286: wheat plants inoculated with appropriate strains. Wheat plants cv. Leningradskaya 6 were grown under hydroponic conditions for 14 days (7 days of exposure to osmotic stress). Bars show ±SEM. Values in columns followed by different letters are significantly different at *p* ≤ 0.05, as determined by Duncan’s multiple-range test.

**Figure 7 plants-13-03381-f007:**
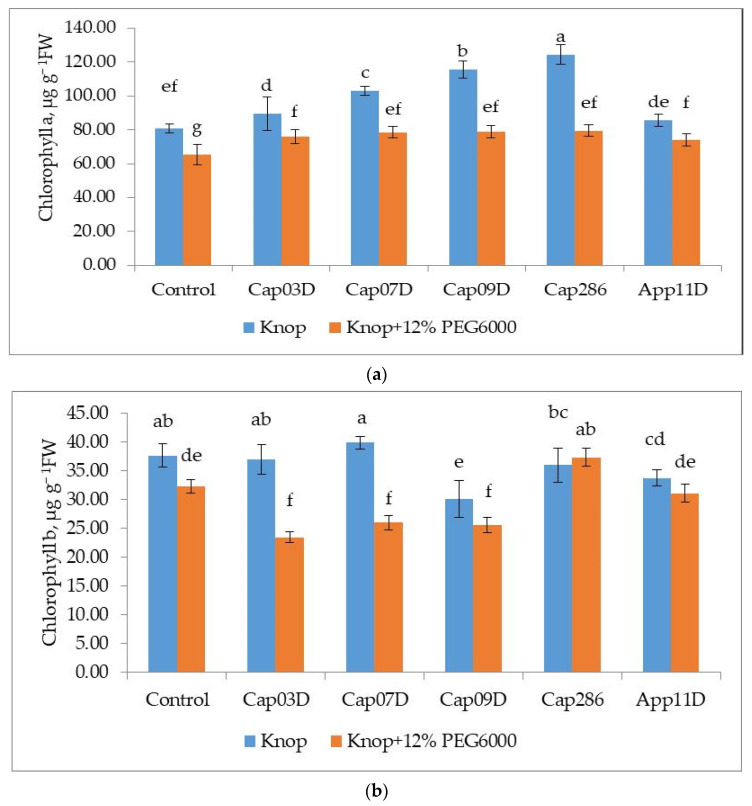
Effect of inoculation with PGPB on chlorophyll a (**a**) and chlorophyll b (**b**) content in wheat leaves cv. Leningradskaya 6, grown under normal and osmotic stress conditions. Notes: Control: non-inoculated wheat plants, growing under control conditions (Knop solution) and under osmotic stress (the addition of 12% PEG6000). Cap03D, Cap07D, Cap09D, App11D, and Cap286: wheat plants inoculated with appropriate strains. Wheat plants cv. Leningradskaya 6 were grown under hydroponic conditions for 14 days (7 days of exposure to osmotic stress). Bars show ±SEM. Values in columns followed by different letters (a–g) are significantly different at *p* ≤ 0.05, as determined by Duncan’s multiple-range test.

**Figure 8 plants-13-03381-f008:**
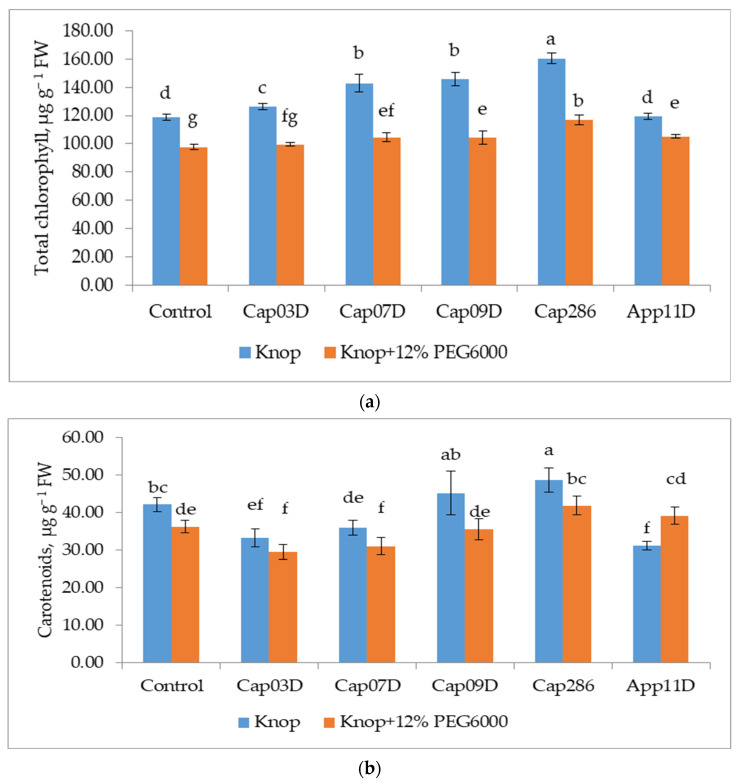
Effect of inoculation with PGPB on total chlorophyll (**a**) and carotenoids (**b**) contents in wheat leaves cv. Leningradskaya 6, grown under normal and osmotic stress conditions. Notes: Control: non-inoculated wheat plants, growing under control conditions (Knop solution) and under osmotic stress (the addition of 12% PEG6000). Cap03D, Cap07D, Cap09D, App11D, and Cap286: wheat plants inoculated with appropriate strains. Wheat plants cv. Leningradskaya 6 were grown under hydroponic conditions for 14 days (7 days of exposure to osmotic stress). Bars show ±SEM. Values in columns followed by different letters (a–g) are significantly different at *p* ≤ 0.05, as determined by Duncan’s multiple-range test.

**Figure 9 plants-13-03381-f009:**
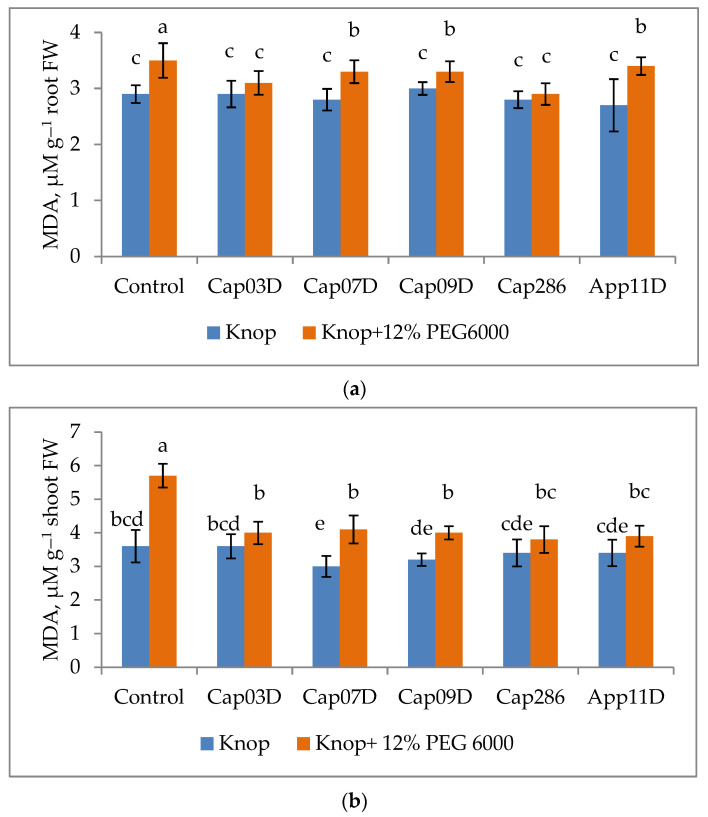
Effect of with PGPB on MDA content in roots (**a**) and shoots (**b**) of wheat cv. Leningradskaya 6, grown under normal and osmotic stress conditions. Notes: Control: non-inoculated wheat plants, grown under normal conditions (Knop) and under osmotic stress (the addition of 12% PEG6000). Cap03D, Cap07D, Cap09D, App11D, and Cap286: wheat plants inoculated with appropriate strains. Wheat plants cv. Leningradskaya 6 were grown under hydroponic conditions for 14 days (7 days of exposure to osmotic stress). Bars with different letters are significantly different at *p* ≤ 0.05, as determined by Duncan’s multiple-range test.

**Figure 10 plants-13-03381-f010:**
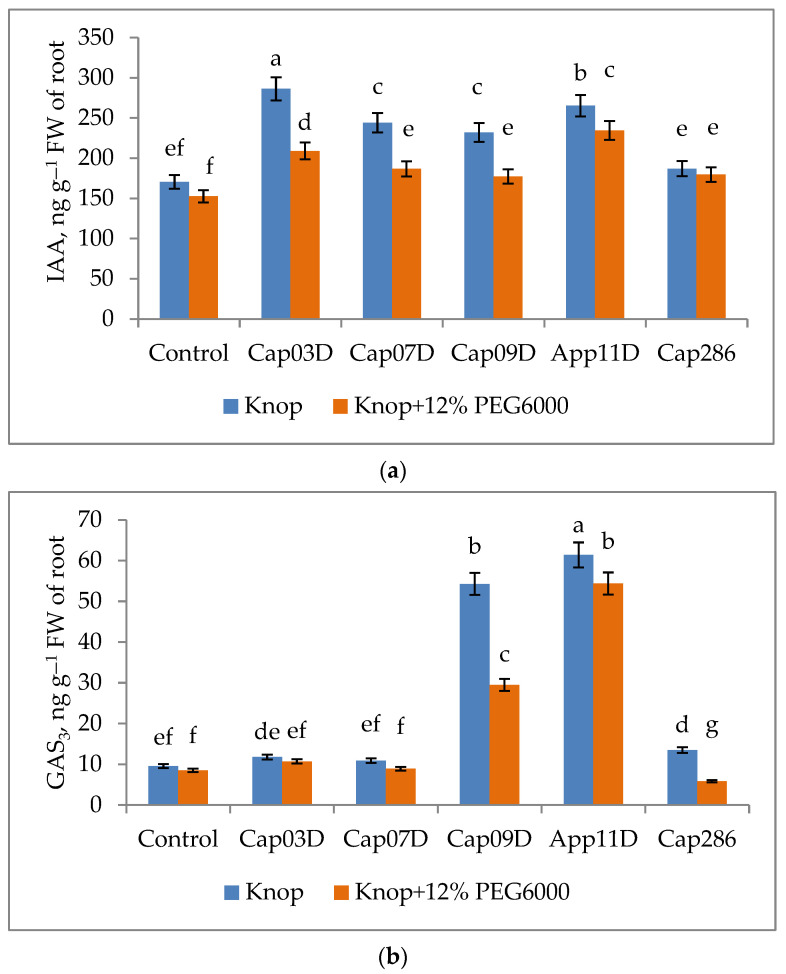
Effect of inoculation with PGPB on concentrations of plant hormones: (**a**) indole-3-acetic acid (IAA); (**b**) gibberellic acid (GAS_3_), and (**c**) abscisic acid (ABA) in roots of wheat plants grown under normal and osmotic stress conditions. Notes: Control: non-inoculated wheat plants, growing under normal conditions (Knop solution) and under osmotic stress (the addition of 12% PEG6000). Cap03D, Cap07D, Cap09D, App11D, and Cap286: wheat plants inoculated with appropriate strains. Bars with different letters are significantly different at *p* ≤ 0.05, as determined by Duncan’s multiple-range test. Wheat plants cv. Leningradskaya 6 were grown under hydroponic conditions for 14 days (7 days of exposure to osmotic stress).

**Table 1 plants-13-03381-t001:** Growth temperature of the PGPB studied.

Strain	5 °C	10 °C	15 °C	28 °C	35 °C	37 °C	40 °C	45 °C	50 °C	55 °C
*Arthrobacter* sp. Cap 03D	−	++	++	++	+	−	−	−	−	−
*Bacillus* sp. Cap 07D	++	+	+++	++	++	++	++	−	−	
*Bacillus* sp. Cap 09D	−	+	++	+++	+++	+++	+++	+++	+++	+
*Paenibacillus* sp. Cap 286	+	+	++	+++	+++	++	+	−	−	−
*Bacillus* sp. App 11D	−	+	+++	+++	+++	+++	+++	+++	+++	−

Notes: +++ active growth, ++ medium growth, + weak growth, and − no growth.

**Table 2 plants-13-03381-t002:** Growth in presence of NaCl.

Bacteria	NaCl, 1%	NaCl, 5%	NaCl, 7%	NaCl, 10%	NaCl, 15%
*Arthrobacter* sp. Cap 03D	++	−	−	−	−
*Bacillus* sp. Cap 07D	++	++	+	−	−
*Bacillus* sp. Cap 09D	++	++	++	++	−
*Bacillus* sp. App 11D	++	++	++	++	−
*Paenibacillus* sp. Cap 286	+	−	−	−	−

Notes: ++ medium growth, + weak growth, and − no growth.

**Table 3 plants-13-03381-t003:** Metabolic activities of PGPB.

Strain	AA	LA	PM	PA	CA	NA	AP	IP	PP	SP
*Arthrobacter* sp. Cap 03D	+	−	−	−	−	−	−	+	−	+
*Bacillus* sp. Cap 07D	+	−	+	+	−	−	+	+	+	+
*Bacillus* sp. Cap 09D	−	+	−	++	−	−	+	+	+	+
*Paenibacillus* sp. Cap 286	++	−	−	+	+	+	−	+	+	+
*Bacillus* sp. App 11D	−	−	−	++	−	−	−	+	+	+

Notes: AA: amylolytic activity; LA: lypolytic activity; PM: phosphate-mobilizing activity; PA: proteolytic activity; CA: cellulolytic activity; NA: nitrogen-fixing activity; AP: (NH4) production; IP: indole production; PP: polysaccharide production; SP: siderophore production. ++ strong activity, + medium and weak activity, and − no activity.

**Table 4 plants-13-03381-t004:** Phytohormones production activity of PGPB.

Phytohormone Production	Cap 03D *Arthrobacter* sp.	Cap 07D *Bacillus* sp.	Cap 09D *Bacillus* sp.	App 11D *Paenibacillus* sp.	Cap 286 *Bacillus* sp.
GAS_3_, ng/mL	29.6 ± 2.19 ^a^	19.8 ± 1.06 ^b^	14.1 ± 2.05 ^c^	17.0 ± 3.05 ^bc^	13.7 ± 2.6 ^c^
IAA, µg/mL	0.37 ± 0.02 ^e^	5.27 ± 1.06 ^a^	1.38 ± 0.06 ^d^	1.82 ± 0.32 ^c^	3.82 ± 0.08 ^b^
ABA, ng/mL	15.9 ± 3.62 ^cd^	37.0 ± 2.80 ^b^	19.3 ± 1.27 ^c^	54.0 ± 4.30 ^a^	11.7 ± 1.25 ^d^

Notes: GAS_3_—gibberellin, IAA—indole acetic acid (content was measured without tryptophan addition), and ABA—abscisic acid. Bars show ±SEM. Values in rows followed by different letters (a–d) are significantly different at *p* ≤ 0.05, as determined by Duncan’s multiple-range test.

**Table 5 plants-13-03381-t005:** Effect of PGPB on wheat cv. Leningradskaya 6 grown under normal and osmotic stress conditions.

Treatment	Length, mm		FW, mg		DW, mg	
	Root	Shoot	Root	Shoot	Root	Shoot
Control	182 ± 6.9 ^c^	360 ± 4.9 ^c^	310 ± 17.2 ^c^	381 ± 19.8 ^c^	22.0 ± 0.6 ^d^	52.2 ± 1.7 ^c^
12PEG	160 ± 6.1 ^d^	305 ± 11.8 ^f^	153 ± 1.45 ^f^	232 ± 14.7 ^e^	10.7 ± 0.8 ^g^	35.3 ± 1.7 ^f^
Cap 03D	209 ± 2.2 ^ab^	373 ±3.9 ^ab^	359 ± 11.3 ^a^	443 ± 23.8 ^a^	26.6 ± 0.7 ^c^	62.0. ± 2.8 ^bc^
12PEG + Cap 03D	175 ± 4.7 ^c^	316 ± 5.7 ^ef^	157 ± 0.64 ^ef^	251 ± 11.9 ^de^	11.9 ± 0.6 ^g^	39.7 ± 1.9 ^e^
Cap 07D	199 ± 4.3 ^b^	375 ± 7.2 ^ab^	342 ± 27.7 ^ab^	403 ± 26.8 ^bc^	29.1 ± 1.1 ^b^	65.3 ± 2.8 ^b^
12PEG + Cap 07D	175 ±8.3 ^c^	331 ± 9.9 ^d^	183 ± 1.76 ^e^	269 ± 14.2 ^d^	16.5 ± 0.8 ^f^	43.1. ± 2.1 ^d^
Cap 09D	211 ± 1.2 ^a^	364 ± 10.0 ^bc^	353± 22.9 ^a^	427 ± 22.6 ^ab^	33.2 ± 1.7 ^a^	77.3 ± 3.4 ^a^
12PEG + Cap 09D	179 ± 5.0 ^c^	327 ± 9.8 d^e^	203 ± 1.02 ^d^	266 ± 13.4 ^d^	19.7 ± 1.0 ^e^	48.9 ± 2.4 ^cd^
App 11D	210 ± 9.7 ^ab^	378 ± 2.7 ^a^	347 ± 22.9 ^ab^	429 ±18.9 ^ab^	31.9 ± 1.1 ^ab^	76.4 ± 3.1 ^a^
12PEG + App 1D	180 ± 9.2 ^c^	362 ± 5.6 ^bc^	196 ± 2.07 ^de^	279 ± 23.0 ^d^	18.8 ± 1.1 ^e^	51.9 ± 2.4 ^c^
Cap 286	209 ± 10.8 ^ab^	368 ± 9.3 ^abc^	360± 26.1 ^a^	419 ±24.5 ^ab^	32.6 ± 2.1 ^a^	76.7 ± 2.0 ^a^
12 PEG + Cap 286	179 ± 2.8 ^c^	327 ± 10.7 ^de^	187 ± 1.02 ^de^	275 ± 20.2 ^d^	18.3 ± 1.0 ^ef^	47.7 ± 1.8 ^cd^

Notes: Control: non-inoculated wheat plants. 12PEG: the addition of 12% PEG6000. Cap 03D, Cap 07D, Cap 09D, App 11D, and Cap 286: wheat plants inoculated with appropriate strains. FW—fresh weight of shoots and roots. Bars show ±SEM. Values in columns followed by different letters (a–f) are significantly different at *p* ≤ 0.05, as determined by Duncan’s multiple-range test. Wheat plants cv. Leningradskaya 6 were grown under hydroponic conditions for 14 days (7 days of exposure to osmotic stress).

**Table 6 plants-13-03381-t006:** Effect of inoculation with PGPB on POX (U s^−1^ g^−1^ FW) and CAT (µM H_2_O_2_ mg^−1^ protein min^−1^) activities of wheat cv. Leningradskaya 6 grown under normal and osmotic stress conditions (7 days).

Treatment	POX in Roots U s^−1^ g^−1^ FW	POX in Shoots U s^−1^ g^−1^ FW	CAT in Roots µM H_2_O_2_ mg^−1^ Protein min^−1^	CAT in Shoots µM H_2_O_2_ mg^−1^ Protein min^−1^
Control	58.94 ± 2.01^de^	31.06 ± 1.41 ^e^	203.82 ± 4.65 ^a^	525.16 ± 12.31 ^a^
12PEG	83.81 ± 3.90 ^a^	53.85 ± 3.76 ^b^	150.9 ± 5.24 ^g^	456.66 ± 19.16 ^b^
Cap 03D	74.15 ± 2.40 ^b^	37.69 ± 3.07 ^d^	173.73 ± 4.60 ^ef^	509.86 ± 11.17 ^a^
Cap 03D-12PEG	77.41 ± 3.78 ^b^	44.28 ± 3.13 ^c^	183.92 ± 3.06 ^bc^	425.37 ± 6.86 ^de^
App 11D	55.95 ± 2.96 ^e^	35.58 ± 1.88 ^d^	179.86 ± 4.48 ^cde^	464.74 ± 7.08 ^b^
App 11D-12PEG	59.51 ± 4.89 ^de^	45.31 ± 3.24 ^c^	188.06 ± 4.47 ^b^	452.72 ± 6.88 ^bc^
Cap 07D	68.8 ± 4.79 ^c^	31.57 ± 1.36 ^e^	175.92 ± 5.04 ^def^	421.43 ± 9.94 ^de^
Cap 07D-12PEG	75.01 ± 2.59 ^b^	72.99 ± 4.63 ^a^	184.31 ± 4.96 ^bc^	422.98 ± 10.02 ^de^
Cap 09D	60.13 ± 2.65 ^de^	28.76 ± 1.59 ^e^	169.58 ± 4.14 ^f^	459.78 ± 16.44 ^b^
Cap 09D-12PEG	64.24 ± 5.49 ^cd^	48.14 ± 2.04 ^c^	182.3 ± 5.15 ^bcd^	436.74 ± 12.42 ^cd^
Cap 286	59.49 ± 4.57 ^de^	48.0 ± 1.5 ^c^	189.89 ± 5.42 ^b^	412.05 ± 8.43 ^e^
Cap 286-12PEG	61.51 ± 2.50 ^de^	56.4 ± 3.3 ^b^	202.27 ± 4.45 ^a^	466.49 ± 10.73 ^b^

Notes: Control: non-inoculated wheat plants, growing under normal conditions; 12 PEG: wheat plants growing under osmotic stress conditions (the addition of 12% PEG6000). Bars show SEM and different letters show a significant difference among treatments at *p* < 0.05 level as determined by Duncan’s multiple-range test. Wheat plants cv. Leningradskaya 6 were grown under hydroponic conditions for 14 days (7 days of exposure to osmotic stress).

## Data Availability

The data presented in this study are available upon request from the corresponding authors.

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
