# Peer review of "PGPB Isolated from Drought-Tolerant Plants Help Wheat Plants to Overcome Osmotic Stress"

_plants, 2024, doi:10.3390/plants13233381_

Round 1

Reviewer 1 Report

Comments and Suggestions for Authors

Rhizosphere microbes promote plant growth and enhance plant stress tolerance, playing an extremely important role in the development of green agriculture. Although the author did not systematically isolate the rhizosphere bacteria from plants in arid regions, a complete analysis was conducted on the characteristics of several strains and their promoting effects on plant growth.

But, the method of PEG treatment needs to be further discussed here. Does PEG have effect on the bacteria?

If microbial agents can be directly applied to economic plants in arid areas, it would be better.

Author Response

Dear reviewer 1,

We are very thankful to Reviewer 1 for valuable comments. We appreciate the time and efforts reviewer 1 dedicated to providing insightful feedback on ways to strengthen our paper. We are fully agreed with reviewer valuable comments and tried to improve the manuscript according his suggestions.

All your comments have been taken into account and corrections have been made. All corrections are marked in yellow.

1Q. But, the method of PEG treatment needs to be further discussed here. Does PEG have effect on the bacteria?

1 (A) Information on the PEG 6000 Treatment Method has been added to L.224-228, 687-691+FigS1, please, see.

2Q. If microbial agents can be directly applied to economic plants in arid areas, it would be better.

2A. Thank you very much for the scientific idea, we will use in further research.

Reviewer 2 Report

Comments and Suggestions for Authors

The paper investigates the role of plant growth-promoting bacteria (PGPB) isolated from drought-tolerant plants in enhancing the drought resistance of wheat (Triticum aestivum L.) under osmotic stress induced by polyethylene glycol (PEG). The authors hypothesize that these bacterial strains contribute to improved growth metrics and biochemical parameters in wheat. The study addresses a critical issue in agriculture—drought stress—especially given the increasing prevalence of drought conditions globally. The exploration of microbial solutions aligns well with sustainable agricultural practices. The use of a hydroponic system to assess the effects of PGPB on wheat growth under controlled conditions allows for clear, reproducible results. The methodology of employing different bacterial strains provides a robust comparative analysis. Results are presented with appropriate statistical analyses, including ANOVA and Duncan’s multiple range test, which enhances the credibility of the findings. The paper effectively connects physiological outcomes (e.g., increased root and shoot biomass, pigment concentrations) with physiological mechanisms (e.g., hormonal modulation), particularly in regard to IAA, ABA, and GAS3 production. So the idea of utilizing PGPB from drought-tolerant plants to improve the resilience of a major crop like wheat is novel and presents potential for practical applications in agro-technology.

However, there are some problems, which must be solved before it is considered for publication. If the following problems are well-addressed, I believe that the paper can be published.

(1) The findings are based on specific bacterial strains (e.g., Bacillus spp. And Paenibacillus sp.) isolated from two drought-resistant plants. The generalizability of these results to other plant species or different PGPB remains uncertain and should be acknowledged.

(2) Although the authors reference various studies on PGPB and drought stress, a more comprehensive discussion on the existing literature would strengthen the manuscript. So, including more recent studies may provide a clearer picture of how their research builds upon or differs from previous work.

(3) While hydroponics provide a controlled environment, field conditions present varying stressors (e.g., soil type, other microorganisms, environmental factors) that could impact PGPB efficacy. It would be beneficial to include or suggest further studies under field conditions for a comprehensive evaluation.

(4) The figures are informative, but some captions lack clarity. More descriptive captions would improve their utility. For example, the captions could more clearly state the significance of the differences represented by the letters.

(5) The paper suggest a pathway for future research into the application of these strains in agricultural practices. However, the paper would benefit from a more detailed discussion of the challenges or considerations that may be faced when implementing these findings in real-world agricultural scenarios.

Conclusion

Overall, the study appears to make a valuable contribution to the understanding of microbial interactions in crop resilience to drought stress. With some refinements and additional context, the paper could effectively communicate its findings and underscore the relevance of PGPB in sustainable agriculture.

In summary, I recommend the authors address the identified weaknesses before publication to ensure a comprehensive and impactful contribution to the field of plant microbiology and agricultural sciences.

Author Response

Dear reviewer 2,

We are very thankful to Reviewer 2 for valuable comments. We appreciate the time and efforts reviewer 2 dedicated to providing insightful feedback on ways to strengthen our paper. We are fully agreed with reviewer valuable comments and tried to improve the manuscript according his suggestions.

All your comments have been taken into account and corrections have been made. All corrections are marked in green.

 (1.) The findings are based on specific bacterial strains (e.g., Bacillus spp. and Paenibacillus sp.) isolated from two drought-resistant plants. The generalizability of these results to other plant species or different PGPB remains uncertain and should be acknowledged.

(1) A. We agree that our findings require further verification. We've corrected our findings. Please, see line 900, 903,905, 909.

 (2) Although the authors reference various studies on PGPB and drought stress, a more comprehensive discussion on the existing literature would strengthen the manuscript. So, including more recent studies may provide a clearer picture of how their research builds upon or differs from previous work

(2)A.  Information has been added. Please see Line 638-650

Salem,2024;  Coutinho et al., 2024 were include in analysis

(3) While hydroponics provide a controlled environment, field conditions present varying stressors (e.g., soil type, other microorganisms, environmental factors) that could impact PGPB efficacy. It would be beneficial to include or suggest further studies under field conditions for a comprehensive evaluation.

(3) A. .  Information has been added. Please, see Line 636-637.

(4).The figures are informative, but some captions lack clarity. More descriptive captions would improve their utility. For example, the captions could more clearly state the significance of the differences represented by the letters.

(4)A. We've added some descriptions in the captions of the Figures. Please, see Fig.9 .

 (5). The paper suggest a pathway for future research into the application of these strains in agricultural practices. However, the paper would benefit from a more detailed discussion of the challenges or considerations that may be faced when implementing these findings in real-world agricultural scenarios.

(5)A. Information has been added. Please, see L.651-662.

Reviewer 3 Report

Comments and Suggestions for Authors

The reviewed paper is an interesting research article, and after revision may be published in the journal. However, before publishing it, I recommend making some changes suggested in the review that will help improve its quality.

I suggest making the following changes for authors:

Line 22, 23 should be „(Alhagi pseudalhagi (M.Bieb.) Fisch.), (Chenopodium album L.), (Triticum aestivum L.)”

Line 105, change “Chenopodium album L.” to “C. album”; line 571 change “Alhagi pseudoalhagi” to “A. pseudalhagi”, please use the abbreviated name of these plants from this point on in the manuscript.

Line 112, should be “Figure 1”

Line 144, should be „Growth temperature…”

In Table 1 there is an empty column between the data for temperatures of 40 and 45oC - please remove it.

Line 166, should be “All strains produced indoles and siderophores”

Line 170, should be “Amylolytic activity”

Line 172, should be “(NH4) production”

Line 183, should be “Figure 3”

Line 184, should be „Bacillus sp. (58.8 mg·L-1)”, please use this unit format throughout the manuscript.

Line 209-212, Please correct this sentences according the data on Figure 1 “The growth-stimulating activity was tested after growing wheat cv. Leningrad. 6 for 4 days (Figure 4). All strains have shown growth-stimulating activity increasing root (by 7.3-18.3%) and shoot growth (by 0.6-8.4 %). The highest growth-stimulating activity was detected in Cap 07D, Cap 09 D and Cap 286 Bacillus spp. strains.”

Please check the correctness of the descriptions regarding the results.

Line 279-281, should be “Paenibacillus sp.”, “Arthrobacter sp.”, “Paenibacillus sp.”, please use italics for Latin names throughout the manuscript.

Statistical analysis for individual characteristics should be separate, e.g. separate for root and stem (Figure 4), and separate for wheat plants growing under control conditions (Knop) and under osmotic stress (Knop+12% PEG6000).

Line 417, should be “abscisic acid (ABA)”

Line 577, should be “the protocol described in Pishchik et al.  [57]”, please use this citation format throughout the manuscript.

Line 584, please use one variety name throughout the manuscript, i.e. "Leningradskaya 6" or "Leningrad.6”

Line 609-610, change “the RDP (https://rdp.cme.msu.edu (accessed on October 10, 2024) [125 and GenBank (https://blast.ncbi.nlm.nih.gov (accessed on October 10, 2024) [126] databases” to “the RDP [125] and GenBank [126] databases.” In the References you should include links to these databases.

Line 618, should be “CaCO3

Line 659, should be „GAS3

Please provide the formulas used to calculate the content of the analyzed parameters, e.g. chlorophylls and carotenoids.

Line 745, 747, 748,  should be “cm3

Line 753, should be “(590 nm)”

Line 761, “this formula “A= 761 D*a*b*c/f*t” should be from a new line, and a legend should be provided below the formula.

Statistical analysis – At what level of significance (p<...) were the differences between the means determined? Whether the normality of the distribution of all data was tested, if so, with what test.

Author Response

Dear reviewer 3,

We are very thankful to Reviewer 3 for valuable comments and corrections in the manuscript. We appreciate the time and efforts reviewer 3 dedicated to providing insightful feedback on ways to strengthen our paper. We are fully agreed with reviewer valuable comments and tried to improve the manuscript according his suggestions.

All your comments have been taken into account and corrections have been made. All corrections are marked in blue.

 (1) Line 22, 23 should be „(Alhagi pseudalhagi (M.Bieb.) Fisch.), (Chenopodium album L.), (Triticum aestivum L.)”

1A.Corrected. Please, see Line 22-23.

(2) Line 105, change “Chenopodium album L.” to “C. album”; line 571 change “Alhagi pseudoalhagi” to “A. pseudalhagi”, please use the abbreviated name of these plants from this point on in the manuscript.

2A. Corrected Please, see Line 104-106.

3) Line 112, should be “Figure 1”

3A. Corrected. Please, see Line112.

(4) Line 144, should be „Growth temperature…

”4.A. Corrected. Please, see Line 145

(5) In Table 1 there is an empty column between the data for temperatures of 40 and 45oC - please remove it.

5A. Done. Please, see Table1.

(6) Line 166, should be “All strains produced indoles and siderophores

6A. Corrected. Please, see Line166-167.

(7) Line 170, should be “Amylolytic activity”

7A. Corrected. Please, see, Line170.

(8) Line 172, should be “(NH4) production”

8A. Corrected. Please, see, Line171-172.

(9) Line 183, should be “Figure 3”

9 A. Corrected. Please, see Line 183.

 (10) Line 184, should be „Bacillus sp. (58.8 mg·L-1)”, please use this unit format throughout the manuscript.

Corrected. Please, see Line184.

(11) 209-212, Please correct this sentences according the data on Figure 1

10A. Corrected. Please, see Line 212-215.

“.The growth-stimulating activity was tested after growing wheat cv. Leningrad. 6 for 4 days (Figure 4). All strains have shown growth-stimulating activity increasing root (by 7.3-18.3%) and shoot growth (by 0.6-8.4 %). The highest growth-stimulating activity was detected in Cap 07D, Cap 09 D and Cap 286 Bacillus spp. strains.”

Please check the correctness of the descriptions regarding the results.

11 A. Corrected. Please, see Line 212-215.

(12) Line 279-281, should be “Paenibacillus sp.”, “Arthrobacter sp.”, “Paenibacillus sp.”, please use italics for Latin names throughout the manuscript.

12 A. Corrected  Please, see Line 276-278.

 (13) Statistical analysis for individual characteristics should be separate, e.g. separate for root and stem (Figure 4), and separate for wheat plants growing under control conditions (Knop) and under osmotic stress (Knop+12% PEG6000).

13A. We corrected the error and recalculated the results. Please, see fig.4

 (14) Line 417, should be “abscisic acid (ABA)”

14A. Corrected. Please, see Line486.

(15 ) Line 577, should be “the protocol described in Pishchik et al.  [57]”, please use this citation format throughout the manuscript.

15A. This citation and other corrected, according MDPI format Please, see Line 683.

(16) Line 584, please use one variety name throughout the manuscript, i.e. "Leningradskaya 6" or "Leningrad.6”

16A. Corrected.

  1. Line 609-610, change “the RDP (https://rdp.cme.msu.edu (accessed on October 10, 2024) [125 and GenBank (https://blast.ncbi.nlm.nih.gov (accessed on October 10, 2024) [126] databases” to “the RDP [125] and GenBank [126] databases.” In the References you should include links to these databases.

17A. We included the links to References

(18) Line 618, should be “CaCO3

18A.Corrected. Please, see Line717.

 (19) Line 659, should be „GAS3

19A. Corrected. Please, see Line758.

(20) Please provide the formulas used to calculate the content of the analyzed parameters, e.g. chlorophylls and carotenoids.

20A. Formulas have been added  Please, see Line 824-830.

(21) Line 745, 747, 748,  should be “cm3

21A. Corrected. Please ,see Line 853,855,856.

  1. Line 753, should be “(590 nm)”

22 A. Corrected. Please, see Line 862.

  1. Line 761, “this formula “A= 761 D*a*b*c/f*t” should be from a new line, and a legend should be provided below the formula.

Corrected  Please,s ee page A, Line 871-878.

24.Statistical analysis – At what level of significance (p<...) were the differences between the means determined? Whether the normality of the distribution of all data was tested, if so, with what test.

We add information in Material and Metods. Please, see Line 889-890.

We used Shapiro-Wilk’s test